# Shear-Mediated Platelet Microparticles Demonstrate Phenotypic Heterogeneity as to Morphology, Receptor Distribution, and Hemostatic Function

**DOI:** 10.3390/ijms24087386

**Published:** 2023-04-17

**Authors:** Yana Roka-Moiia, Kaitlyn R. Ammann, Samuel Miller-Gutierrez, Jawaad Sheriff, Danny Bluestein, Joseph E. Italiano, Robert C. Flaumenhaft, Marvin J. Slepian

**Affiliations:** 1Sarver Heart Center, Departments of Medicine and Biomedical Engineering, University of Arizona, 1501 N Campbell Ave, Building 201E, Room 6139, Tucson, AZ 85724, USA; rokamoiia@email.arizona.edu (Y.R.-M.);; 2Department of Biomedical Engineering, Stony Brook University, Stony Brook, NY 11794, USA; 3Boston Children’s Hospital, Harvard Medical School, Boston, MA 02115, USA; 4Beth Israel Deaconess Medical Center, Boston, MA 02215, USA

**Keywords:** platelet-derived microparticles, shear stress, adhesion receptors, agonist receptors, cardiovascular therapeutic devices, bleeding, thrombosis

## Abstract

Implantable Cardiovascular Therapeutic Devices (CTD), while lifesaving, impart supraphysiologic shear stress to platelets, resulting in thrombotic and bleeding coagulopathy. We previously demonstrated that shear-mediated platelet dysfunction is associated with downregulation of platelet GPIb-IX-V and αIIbβ3 receptors via generation of Platelet-Derived MicroParticles (PDMPs). Here, we test the hypothesis that sheared PDMPs manifest phenotypical heterogeneity of morphology and receptor surface expression and modulate platelet hemostatic function. Human gel-filtered platelets were exposed to continuous shear stress. Alterations of platelet morphology were visualized using transmission electron microscopy. Surface expression of platelet receptors and PDMP generation were quantified by flow cytometry. Thrombin generation was quantified spectrophotometrically, and platelet aggregation was measured by optical aggregometry. Shear stress promotes notable alterations in platelet morphology and ejection of distinctive types of PDMPs. Shear-mediated microvesiculation is associated with the remodeling of platelet receptors, with PDMPs expressing significantly higher levels of adhesion receptors (α_IIb_β_3_, GPIX, PECAM-1, P-selectin, and PSGL-1) and agonist receptors (P_2_Y_12_ and PAR1). Sheared PDMPs promote thrombin generation and inhibit platelet aggregation induced by collagen and ADP. Sheared PDMPs demonstrate phenotypic heterogeneity as to morphology and defined patterns of surface receptors and impose a bidirectional effect on platelet hemostatic function. PDMP heterogeneity suggests that a range of mechanisms are operative in the microvesiculation process, contributing to CTD coagulopathy and posing opportunities for therapeutic manipulation.

## 1. Introduction

Implantable Cardiovascular Therapeutic Devices (CTD) have become the mainstay of advanced therapy for a broad range of cardiovascular diseases. In recent years, we have witnessed growth in the implantation of stents [1], percutaneous heart valves [2], and mechanical circulatory support, with placement of percutaneous or surgical ventricular assist devices for advanced and end-stage heart failure [3,4]. While these devices have demonstrated hemodynamic efficacy, implantation can negatively affect hemocompatibility [5]. To enhance device hemocompatibility, anticoagulant and antiplatelet agents are administered clinically [6]. While pharmacologic agents have demonstrated efficacy in limiting stent thrombosis [7], for other devices, it is increasingly evident that drugs have limited efficacy and contribute to adverse bleeding events [8,9]. Presently, there is a movement to curtail the use of antithrombotic agents to reduce CTD-related bleeding [10]. 

Hemocompatibility of implanted devices is mediated by a delicate balance of bio-material considerations, thrombogenicity and inflammatory potential of blood, and flow-related disturbances imparted by abnormal shear stresses [11,12,13]. In recent years, device hemocompatibility has improved; however, due to design-dictated flow regimes and geometries, supraphysiologic shear stress is still imparted to blood cells and proteins as they traverse devices, resulting in biological consequences [14,15]. We and others have reported on the role of elevated shear stress as a driver of device-related coagulopathy with both prothrombotic and bleeding features [16,17,18]. Shear-mediated platelet activation via mechanotransductive and mechanodestructive means has been demonstrated to drive thrombosis [19,20,21]. Concomitantly, continuous exposure to hypershear results in platelet dysfunction and subsequent device-related bleeding [22,23].

Platelets are small anucleate blood cells essential for maintaining hemostasis. Platelets operate in delicate equipoise between thrombosis and bleeding, biochemically and physiologically controlled to avoid inadvertent thrombosis or bleeding with dire systemic consequences. When exposed to biochemical and mechanical stimuli, platelets undergo activation, rapid shape change, release of secretory granules, and ultimately aggregate to form primary thrombi. In their ability to sense and interact with the extracellular environment, platelets rely on two classes of GlycoProtein (GP) receptors: *adhesion* receptors and *agonist-evoked* receptors. *Adhesion receptors* are abundantly expressed on platelet surfaces, either constitutively (GPIb-V-IX, integrin α_IIb_β_3_, GPVI, PECAM-1) or “on-demand”, as a result of activation (P-selectin, PSGL-1) [24]. During platelet aggregation, GPIb-V-IX primarily binds to von Willebrand factor (vWF), and integrin α_IIb_β_3_ binds to fibrinogen and vWF, whereas GPVI interacts with collagen. PECAM-1, P-selectin, and PSGL-1 are responsible for platelets binding to leukocytes and vascular endothelium [24]. The most well-known and therapeutically relevant *agonist receptors* are ADP-evoked P_2_Y_12_ and thrombin-evoked PAR1 receptors. These receptors are scarcely, yet constitutively, expressed on platelets and are targets for common antiplatelet agents. The surface expression of GP receptors is tightly regulated to limit spontaneous platelet adhesion and activation upon interaction with plasma proteins and other cells. Enzymatic shedding of a ligand-binding ectodomain is a well-documented regulatory mechanism of platelet receptor surface expression [25,26]. Both continuous and activation-induced types of enzymatic shedding have been described for GPIb, GPVI, P-selectin, and PECAM-1 [25,26,27]. An alternative mechanism of downregulation of receptor surface expression is restructuring of the entire platelet membrane as a result of microvesiculation and microparticle generation [23,28]. 

Platelet-Derived MicroParticles (PDMPs) are submicron, membrane-enclosed, extracellular vesicles released from platelets during their activation by strong agonists, aging, and apoptosis [29,30,31]. Platelets and megakaryocytes are the primary sources of microvesicles circulating in blood, with up to 90% carrying canonical platelet antigens—CD41, CD42b, GPVI, and externalized phosphatidylserine [29]. Detailed mechanisms of PDMP formation have been described emphasizing the importance of high intracellular calcium, actin cytoskeleton rearrangement, and apoptotic machinery [32]. PDMP proteomics have revealed several types of microvesicles differing in size, protein composition, and effect on clotting and endothelial cell phenotype [33]. PDMPs not only provide an additional surface for activation of the coagulation cascade and thrombin generation but also deliver receptors, nucleic acids, signaling and pro-inflammatory molecules to other cells, serving as a specialized mechanism of intravascular and extravascular communication. Our research group and others have demonstrated that extended exposure to device-related shear stress promotes downregulation of platelet adhesion receptors and is associated with extensive microparticle generation in vitro and in vivo [23,34,35]. Elevated levels of circulating microparticles were more pronounced in device-implanted patients experiencing post-implantation bleeding as opposed to non-bleeders [36,37]. For other CTDs, elevated levels of microparticles were also associated with hemostatic dysfunction, inflammation, and adverse events [38]. Despite the significance of PDMPs in the pathophysiology and diagnosis of device-related coagulopathy, their detailed phenotypic characterization and modulatory effect on platelet hemostatic function have not been investigated.

Here, we hypothesized that (1) exposure to supraphysiologic shear stress promotes alterations of platelet morphology and generation of microparticles; (2) shear-mediated PDMPs manifest phenotypic heterogeneity of their morphology and surface expression of platelet *adhesion* receptors and *agonist* receptors; and (3) shear-mediated PDMPs modulate platelet hemostatic function. Specifically, we examined the effect of shear stress accumulation on the morphology of platelets and PDMPs using Transmission Electron Microscopy (TEM). We defined the effect of shear stress on the distribution of adhesion- and agonist receptors on platelets and microparticles using multi-colored flow cytometry. Finally, we tested the effect of PDMPs on platelet hemostatic function, i.e., thrombin generation and aggregation. 

## 2. Results

### 2.1. Shear-Induced Alterations of Platelet Morphology and Microparticle Generation

TEM examination of the *low-speed fraction* revealed morphological details of resting, sheared, and sonicated platelets (Figure 1A–D). Resting platelets were generally round or ellipsoidal with smooth contours and continuous membranes, containing uniform cytoplasm with multiple granules (Figure 1A). With increasing shear, platelet morphology remained largely intact through exposure to 30 dyne/cm^2^ (data not shown). At 50 dyne/cm^2^ shear, a moderate platelet size alteration, pseudopod extension, and microparticle formation was detected (Figure 1B); at 70 dyne/cm^2^ shear, significant alteration of platelet morphology was widespread (Figure 1C). The majority of platelets had significantly altered morphology—being either round with pale, depleted cytoplasm or fragmented with noticeable microvesiculation and emerging free PDMPs. Other platelets appeared smaller, with signs of pseudopod formation. Sonication, as a form of mechanical activation via acoustic cavitation, disrupted platelets with residual ghost-like structures and free PDMPs (Figure 1D).

The PDMP-rich fraction obtained by *high-speed* centrifugation of resting platelets contained intact platelets and platelet fragments along with trace PDMPs (Figure 1E–H). Rare cigar-shaped platelet fragments (100 × 800 nm) were visible, typically devoid of granules with cytoplasmic density similar to intact platelets. A shear stress increase to 50 dyne/cm^2^ led to a mild increase in the number of platelet fragments and PDMPs (Figure 1F). At 70 dyne/cm^2^ shear, a significant alteration in platelet morphology and an increase in PDMPs was evident (Figure 1G). Intact platelets were outnumbered by platelet remnants with absent granules and depleted cytoplasmic density. Following sonication, no intact platelets were observed (Figure 1H); sonicated PDMPs appeared as circular, ellipsoidal, or amorphous particles with a range of densities, some similar to platelet granules. 

The heterogeneity of platelet and PDMP morphologies after 70 dyne/cm^2^ shear stress exposure is depicted in Figure 2A. Platelet membrane thinning and fragmentation were evident (Figure 2B,C). Two types of PDMPs were noted: (1) large irregular particles, likely platelet fragments, as shown in Figure 2A–G; and (2) small “granule-like” circular or ellipsoidal particles with evident contrast density, or devoid of contents, as best shown in Figure 2F,G. Platelet granules were proximate to the plasma membrane, with apparent budding (Figure 2E).

The number of microparticles generated by shear stress and sonication was also assessed using TEM. Two broad groups of PDMPs were analyzed: small particles (150–500 nm) and large particles (500–1000 nm). Intact platelets and debris were excluded from the analysis. An evident increase in the small particle population occurred following platelet exposure to high shear stress (Appendix A): 113 + 3 vs. 22 ± 1 particles/field in 70 dyne/cm^2^ shear and no shear, respectively. Sonication resulted in even greater PDMP generation. For large particles, a minor increase was observed at 50 and 70 dyne/cm^2^ shear, as compared with unsheared platelets (Appendix A). After low shear and sonication, no large particles were evident.

### 2.2. αIIbβ3 and GPIX Distribution on Platelets and Platelet-Derived Microparticles

Platelet exposure to increasing shear stress and sonication resulted in a significant increase in PDMPs and a decrease in platelet populations. Typical scatter plots of events captured by flow cytometry are shown in Figure 3. The CD41+ PDMP population significantly increased with shear magnitude (Figure 3B–E, gate “Microparticles”); sonication resulted in dramatic microvesiculation (Figure 3F). A significant increase in both CD41+ and CD42a+ PDMP populations was observed after 30 dyne/cm^2^ shear; at 70 dyne/cm^2^ shear, CD41+ and CD42a+ PDMPs accounted for 9.1 ± 1.2% and 5.7 ± 0.4% of all events, respectively (Figure 4A,B). Sonication generated an even greater number of CD41+ and CD42a+ PDMPs. Both CD41+ and CD42a+ *platelet* populations decreased gradually with increasing shear magnitude (Figure 4C,D). Sonication resulted in an even greater decrease in platelet count.

Analyzing receptor distribution on platelet and microparticle surfaces, we (1) monitored the alteration of receptor density across all shear stress conditions and (2) compared the receptor density on platelets and microparticles. Shear stress did not largely affect CD41 and CD42a fluorescence intensity (Appendix A) and platelet size (Appendix A). The arbitrary density of α_IIb_β_3_ and GPIX receptors on platelets was not significantly altered (Appendix A, “Shear stress”). In contrast, sonication resulted in a decrease in platelet size and fluorescence intensity for both CD41 and CD42a, thus rendering a significant decrease in receptor arbitrary density, particularly for α_IIb_β_3_. Similarly, microparticle size and fluorescence intensity remained unchanged across shear conditions for both receptors, while sonication generated smaller PDMPs with significantly lower receptor density (Appendix A, “Sonication”). 

Further, the arbitrary density of α_IIb_β_3_ and GPIX on sheared PDMPs was significantly higher than on platelets across all shear conditions (Figure 4E,F, “Shear stress”). After 70 dyne/cm^2^ shear, CD41+ PDMPs expressed 2.3-fold higher receptor density while CD42a+ PDMPs showed 6-fold higher receptor density than platelets. In contrast, sonicated PDMPs demonstrated similar or barely elevated receptor density as compared with sonicated platelets (Figure 4E,F, “Sonication”).

### 2.3. GPVI and PECAM-1 Distribution on Platelets and Platelet-Derived Microparticles

The number of GPVI+ PDMPs was not elevated, even with the highest levels of shear applied (Figure 5A), while the PECAM-1+ PDMP population increased gradually, accounting for 6.2 ± 0.6% after 70 dyne/cm^2^ shear (Figure 5B). Sonication generated significantly greater numbers of GPVI+ and PECAM-1+ microparticles, accounting for 6.8 ± 1.2% and 18.9 ± 0.6% of events, respectively. The number of platelets expressing GPVI and PECAM-1 steeply declined with the increased magnitude of shear, reaching 66.4 ± 3.4% and 84.0 ± 2.2% after 70 dyne/cm^2^ shear (Figure 5C,D). Sonication resulted in a significant drop in both platelet populations. 

Analyzing the alteration of platelet size and receptor density on GPVI+ platelets following shear exposure, we did not detect a decrease in platelet size, yet fluorescence intensity steeply declined with increasing shear stress (Appendix A). The GPVI arbitrary density on platelets significantly decreased with increased shear (Appendix A). Similarly, shear stress resulted in a significant decline in PECAM-1 arbitrary density on platelets (Appendix A). Sonication resulted in a significant decrease in platelet size and a 20–25% decrease in GPVI and PECAM-1 arbitrary density on platelets (Appendix A, “Sonication”). Shear-mediated microparticles were similar in size across all shear conditions and were slightly larger than those generated by sonication (Appendix A) while expressing significantly higher levels of GPVI and PECAM-1 (Appendix A).

When comparing GPVI and PECAM-1 arbitrary density on platelets and microparticles, we found that sheared PDMPs expressed slightly higher levels of GPVI and CD31 on their surface than sheared platelets (Figure 5E,F). After 70 dyne/cm^2^ shear, the GPVI and PECAM-1 arbitrary density on PDMPs was 1.8-fold higher than on platelets. Microparticles generated by sonication demonstrated very similar receptor levels to those on sonicated platelets.

### 2.4. P-Selectin and PSGL1 Distribution on Platelets and Platelet-Derived Microparticles

The number of P-selectin+ and PSGL1+ PDMPs increased slightly following platelet exposure to shear stress (Figure 6A,B). Sonication resulted in a significant increase in both PDMP populations up to 7.2 ± 0.5% for P-selectin+ and 6.2 ± 0.3% for PSGL1+ PDMPs. Baseline levels of both P-selectin and PSGL1 expressed on platelets were very low, as indicated by the low number of CD62P+ and PSGL1+ platelet populations (Figure 6C,D: “No shear”). Exposure to 70 dyne/cm^2^ shear resulted in a minor increase in P-selectin+ platelets, yet the number of PSGL1+ platelets remained low across all shear magnitudes (Figure 6C,D: “Shear stress”). Sonication resulted in a significant decrease in P-selectin+ and PSGL1+ platelet populations.

Analyzing alterations of platelet size and receptor arbitrary density, we noticed that both high shear and sonication resulted in significant shrinkage of P-selectin+ platelets (Appendix A). The P-selectin density on platelets and PDMPs gradually increased with shear stress magnitude, indicating ongoing degranulation. Sonication resulted in a further increase in P-selectin levels on platelets and PDMPs (Appendix A). Baseline PSGL1 surface expression on platelets was extremely low and remained unchanged across all shear conditions. Interestingly, sonication resulted in a 2-fold increase in PSGL1 density on platelets but not on PDMPs (Appendix A). 

While comparing the receptor arbitrary density on platelets and microparticles, we found that PDMPs expressed significantly higher P-selectin and PSGL1 levels than platelets (Figure 6E,F). Specifically, the arbitrary density of P-selectin and PSGL1 on shear-generated PDMPs was 7 to 10-fold higher than on sheared platelets. While sonicated PDMPs were less densely populated, they still exhibited 4 to 7-fold higher receptor density than platelets. 

### 2.5. P_2_Y_12_ and PAR1 Distribution on Platelets and Platelet-Derived Microparticles

Both sheared and sonicated PDMPs expressed increased levels of P_2_Y_12_ and PAR1 receptors (Figure 7A,B). Both P_2_Y_12_+ and PAR1+ PDMP populations gradually increased with shear magnitude, reaching 3.5 ± 0.6% and 4.3 ± 0.5% after 70 dyne/cm^2^ shear, respectively. Sonication resulted in an even further increase in PDMP populations: 9.6 ± 1.7% and 14.8 ± 1.6% for P_2_Y_12_+ and PAR1+ PDMPs, respectively. Baseline levels of P_2_Y_12_ and PAR1 on platelets were extremely low, with not all platelets expressing detectable levels of these markers. Only 74.7 ± 1.7% of CD41+ platelets were positive for P_2_Y_12_, and 74.0 ± 1.1% of CD41+ platelets were positive for PAR1. Exposure to shear resulted in a minor decrease in both P_2_Y_12_+ and PAR1+ platelet populations even following 30 dyne/cm^2^ shear, reaching 63.3 ± 1.7% and 60.7 ± 1.6% following 70 dyne/cm^2^ shear, respectively. Sonication led to a 3-fold decrease in receptor-positive platelet populations (Figure 7C,D, “Sonication”). 

Shear stress did not significantly affect platelet size, while sonication rendered significantly smaller platelets (Appendix A); the PDMP size varied between shear and sonication (Appendix A). Baseline P_2_Y_12_ density was extremely low and increased slightly following 70 dyne/cm^2^ shear exposure (Appendix A), while PAR1 density on platelets tended to decrease with the increasing shear stress magnitude (Appendix A). Sonication resulted in a nearly 2-fold increase in receptor arbitrary density on platelets but not on PDMPs (Appendix A, “Sonication”). 

Comparing receptor density on sheared platelets and PDMPs, we found that PDMPs expressed nearly 10-fold higher levels of P_2_Y_12_ receptor and 7-fold higher levels of PAR1 (Figure 7E,F). The PDMP populations generated by sonication were even further enriched with agonist receptors, expressing 6.5-fold higher levels of P_2_Y_12_ and 6.3-fold higher levels of PAR1 than sonicated platelets. 

### 2.6. Procoagulant Activity of Platelets and Platelet-Derived Microparticles 

The number of platelets and PDMPs binding annexin V significantly increased with shear stress magnitude (Figure 8A,B). After sonication, the number of annexin V+ platelets was significantly lower than after 70 dyne/cm^2^ shear. However, sonication rendered a 3-fold higher number of annexin V+ PDMPs (Figure 8B). Analyzing the density of annexin V binding on platelets, we found that sheared platelets bound significantly higher levels of annexin V than sonicated counterparts: 4.1 ± 0.0 AU vs. 1.4 ± 0.1 AU for 70 dyne/cm^2^ shear and sonication, respectively. Similarly, sheared PDMPs exhibited 6.2-fold higher levels of annexin V than sonicated PDMPs (Figure 8C). The arbitrary density of annexin V binding on sheared PDMPs was 3-fold higher than on sheared platelets (Figure 8C), while PDMPs generated by sonication bound the same levels of annexin V as sonicated platelets (Figure 8C). 

Platelet exposure to shear stress and sonication promoted thrombin generation, with a resulting thrombin generation rate of 37.9 ± 2.9 min^−1^ and 94.9 ± 16.7 min^−1^, respectively. Sheared PDMPs showed mild prothrombotic activity when incubated alone or with resting platelets (Figure 8D). Yet, sonicated PDMPs rendered a sharp increase in thrombin generation when added alone or with resting platelets, with the resulting thrombin generation rate of 42.7 ± 19.9 min^−1^ and 88.4 ± 2.9 min^−1^, respectively, as compared with 8.2 ± 3.8 min^−1^ in the control group.

### 2.7. The Effect of Platelet-Derived Microparticles on Platelet Aggregation

All biochemical agonists induced platelet aggregation, though the aggregation amplitude and area under the curve varied significantly. ADP and collagen were nearly 10-fold as effective in stimulating aggregation as TRAP-6 (Figure 9). PDMP-rich fractions added to PRP did not promote platelet aggregation in the absence of aggregation agonists (Appendix A). The extent of the modulatory effect of the two PDMP-rich fractions differed for all agonists tested. Platelet aggregation induced by collagen was significantly inhibited by sheared PDMPs, while sonicated PDMPs showed a less prominent inhibitory effect (Figure 9A,B). In the presence of sheared and sonicated PDMPs, collagen-induced platelet aggregation reached only 64.3% and 78.6% of its control level. ADP-induced aggregation was more resilient to inhibition with PDMPs; however, sheared PDMPs demonstrated more potent inhibition than those generated by sonication (Figure 9A,C). PDMP-rich fractions did not significantly alter TRAP-6-mediated platelet aggregation. Although, both sheared and sonicated PDMPs tended to decrease aggregation as compared with the control group (Figure 9A,D).

## 3. Discussion

Persistent exposure to mechanical stresses imparted by CTDs promotes platelet dysfunction and device-related coagulopathy manifested clinically as bleeding and thrombotic adverse events. Our group and others have demonstrated that shear-mediated platelet dysfunction is associated with pro-apoptotic behavior [17,35], decreased platelet aggregation [23,39], and extensive microvesiculation [23,34,35]. Elevated levels of circulated microparticles are associated with adverse events and reduced long-term efficacy of CTD therapy. Here, we defined and characterized the phenotype and functional significance of shear-generated PDMPs. We demonstrated that exposure to continuous high shear stress resulted in notable alterations in platelet morphology and release of PDMPs with varying morphologies and internal characteristics. We found that shear-mediated microvesiculation was associated with differential remodeling of the surface receptors on platelets and PDMPs, with several distinctive patterns emerging. (1) For α_IIb_β_3_ and GPIX, the arbitrary density on sheared platelets was largely unaltered; yet all sheared PDMPs showed a 2 to 6-fold higher density of these receptors. (2) For PECAM-1 and GPVI, the arbitrary density on platelets and PDMPs gradually decreased with increasing shear magnitude; a small population of sheared PDMPs was PECAM-1+, but no GPVI+ PDMPs were released with shear exposure. (3) P-selectin, but not PSGL1, arbitrary density increased slightly on sheared platelets and PDMPs; a small population of shear-mediated PDMPs was P-selectin+ and PSGL1+, expressing significantly higher levels of these receptors than platelets. (4) The arbitrary density of P_2_Y_12_ and PAR1 receptors slightly increased on sheared platelets, and only a minor population of shear-generated PDMPs was positive for P_2_Y_12_ and PAR1, expressing much higher levels of receptors than sheared platelets. (5) PDMPs generated by shear stress had a bidirectional effect on platelet hemostatic function, exhibiting a procoagulant surface promoting thrombin generation and simultaneously inhibiting agonist-induced platelet aggregation.

Microvesiculation with resultant PDMP formation occurs when platelets undergo hyper-activation associated with a persistent increase in intracellular calcium or enter terminal stages of apoptosis. We and others have shown that platelet exposure to device-related shear stress also results in generation of PDMPs [23,34,35,40]. Here, we demonstrated that platelet exposure to continuous shear stress in free flow is accompanied by notable alterations in platelet morphology, quantitative and qualitative alterations of organelles, and release of extracellular vesicles ranging from microparticles to exosomes. Following exposure to moderate continuous shear stress, platelets assumed an asymmetrical shape with multiple filopodia extensions, which drastically differed from the round or discoid shape of resting platelets (Figure 1 and Figure 2). The number of secretory granules greatly decreased, with the remaining granules grouped together and colocalized with the open canalicular system, indicating ongoing degranulation [41,42]. It was previously reported that similar morphological changes accompanied PDMP formation in response to activation by strong biochemical agonists [43]. Microvesiculation induced by thrombin and calcium ionophore was associated with filopodia formation, dilatation of the open canalicular system, degranulation, and defragmentation [43,44,45]. Our findings are in agreement with previous studies of increased levels of shear. A shear rate of 10,000 s^−1^ (~100 dyne/cm^2^) was shown to significantly increase PDMP production in the presence of GPIbα-vWF interaction, with drop-like or spherical particles formed from detached membrane tethers extending from adhered platelets [46]. Similarly, an increased generation of PDMPs was demonstrated when whole blood was exposed to prolonged shear stress ex vivo in extracorporeal membrane oxygenation [47]. 

We observed that exposure to supraphysiologic shear resulted in drastic alterations of platelet morphology. Many platelets resembled a ghost-like appearance with low-contrast cytoplasm occupied by numerous membrane-enclosed vesicles (Figure 2B,C). These vesicles are likely pro-PDMPs or multivesicular PDMPs similar to those reported for thrombin-induced activation [43]. Sheared PDMPs were numerous, substantially varying in size, density, and morphology. The morphological heterogeneity of PDMPs generated by shear and sonication suggests that a variety of mechanisms are operative in PDMP formation. Precedent for a range of operative mechanisms and morphological heterogeneity of PDMPs exists with strong biochemical agonists. It has been reported that thrombin and calcium ionophore promote formation of four ultrastructurally different types of microvesicles: single membrane-enclosed vesicles, multilayer vesicles, multivesicular particles consisting of several vesicles (10–15), and particles with dense contents including cytoplasm and organelles of parental cells [43]. It has been suggested that these four PDMP types are formed via differing mechanisms, including membrane invagination following budding; formation at the end of pseudopodia; degranulation involving the open canalicular system; or platelet fragmentation. While these mechanisms may be operative in shear-mediated PDMP generation, the morphological observations in our study are also consistent with a previously proposed mechanodestruction mechanism wherein shear stress imparts direct mechanical damage to the platelets, resulting in enhanced membrane porogenicity, rapid influx of calcium and other ions, and ultimate membrane fragmentation generating PDMPs [16]. The mechanodestruction mechanism is supported by prior modeling studies of membrane and whole-cell shear stress accumulation [48,49,50,51] and is consistent with our TEM observations. Our ultrastructural study indicates that (1) sheared platelets undergo shape change with cytoplasmic extrusion and pseudopodia extension; (2) sheared platelets are engaged in extensive degranulation; (3) not all platelets are actively involved in PDMP generation, with some platelets remaining intact, while others form ghost-like structures packed with pro-PDMPs; and (4) shear-mediated PDMPs vary in size, density, and ultrastructural features.

Platelet surface receptors are responsible for sensing biochemical and mechanical signals from the intravascular environment, transducing these signals intracellularly to initiate a functional response. *Adhesion receptors* are crucial for platelet adhesion to vascular walls and aggregation to one another. Platelet maturation is associated with increased basal surface expression, while platelet aging and apoptosis result in downregulation of receptor surface expression. Enzymatic shedding and microvesiculation have been identified as two primary mechanisms responsible for downregulation of platelet surface receptors. Our research group was among the first to recognize that platelet exposure to continuous shear stress of increased magnitude and duration promotes downregulation of GPIb, α_IIb_β_3_, and P-selectin surface expression via generation of PDMPs carrying increased levels of these receptors [23,52]. 

Analyzing PDMPs carrying integrin α_IIb_β_3_ and GPIX on their surfaces, we found that continuous shear exposure up to 70 dyne/cm^2^ induced a 5-fold increase in both populations of circulating PDMPs, though the GPIX+ PDMPs were less numerous than their α_IIb_β_3_+ counterparts. A slight decrease in CD41+ and CD42a+ platelets was also seen, likely associated with platelet disintegration accompanying PDMP generation, as reflected in our TEM observations. Shear-mediated PDMPs were increasingly decorated with both α_IIb_β_3_ and GPIX (Figure 4E,F). These findings are in agreement with our previous report, where we first demonstrated the shear-mediated generation of PDMPs carrying increased levels of platelet receptors α_IIb_β_3_ and GPIb following exposure to continuous but not pulsatile shear of increased magnitudes [23]. Interestingly, the surface density of GPIX, but not α_IIb_β_3_, on sheared PDMPs was slightly elevated with increasing shear magnitude. Upregulation of GPIX expression on PDMPs may result from redistribution of internal GPIX from platelet granules following their fusion with platelet plasma membranes or open canalicular systems during granule exocytosis. Redistribution of GPIb and GPIX between platelet surfaces and internal membranes was previously demonstrated during platelet stimulation with strong biochemical agonists [53]. Alternatively, the GPIX-enriched PDMP population may originate from platelet internal granules carrying significant amounts of GPIb-IX [54,55]. Platelet sonication, used as a positive control for platelet mechanodestruction, resulted in platelet disintegration as indicated by a significant platelet count drop and generation of increased numbers of PDMP-sized bodies, carrying both α_IIb_β_3_ and GPIX receptors. However, the surface density of α_IIb_β_3_ and GPIX on microvesicles generated by sonication remained the same or comparable to the receptor density on platelets (Figure 4E,F). This observation strongly suggests that sonication nonspecifically destroys platelets, while shear stress induces a specific platelet response associated with selective redistribution of platelet receptors from platelets to PDMPs and is likely accompanied by platelet degranulation.

Platelet GPVI and PECAM-1 receptors, while not as abundant as α_IIb_β_3_ and GPIb-IX-V, play a prominent role in platelet rolling and adhesion to vascular endothelium, regulation of primary hemostasis, and thrombus formation. GPVI is regarded as a potent signaling receptor for collagen-evoked platelet activation and microparticle release when co-stimulated with other biochemical agonists, i.e., thrombin and ADP [56,57]. We found that, after shear, surface expression of GPVI and PECAM-1 on platelets significantly decreased, evident from the decrease in GPVI and PECAM-1 arbitrary density. Interestingly, sheared PDMPs did not express GPVI, and only a small population presented PECAM-1 on their surface (Figure 5A,B). Proteolytic shedding of both GPVI and PECAM-1 associated with activation, apoptosis, and exposure to hypershear stress has been documented for both platelets and endothelial cells [27,58]. It was previously shown that microvesicles released following platelet activation by biochemical agonists also contain PECAM-1, along with integrins α_IIb_β_3_ and β_1_. Our observations suggest that, following shear exposure, both GPVI and PECAM-1 are downregulated on platelets, while PECAM-1, but not GPVI, is found on shear-mediated microparticles where it is also downregulated (Figure 5F). In contrast, sonication resulted in a massive generation of PECAM-1+ and GPVI+ PDMPs and a corresponding decrease in platelet populations carrying those receptors. Following sonication, the surface density of these receptors was virtually the same on PDMPs and platelets (Figure 5E,F), strongly suggesting that sonication induced random platelet fragmentation. 

In resting platelets, P-selectin is located within α-granule membranes. Upon activation and degranulation, P-selectin is translocated onto the platelet surface [59]. P-selectin undergoes rapid “shutdown” by the shedding of a soluble fragment (sP-selectin) from the platelet surface or by internalization [59]. The reported level of PSGL-1 on unstimulated platelets is very low, increasing following platelet activation with thrombin, and gradually decreasing as platelets age [60]. We also found that unstimulated platelets expressed a very low level of both P-selectin and PSGL-1. The surface density of P-selectin, but not PSGL-1, mildly increased with increasing shear, indicating degranulation. A small population of sheared PDMPs expressed increased levels of both P-selectin and PSGL-1 (Figure 6E,F). The mild increase in P-selectin on sheared platelets might be indicative of low levels of secretory activity stimulated by shear. Yet, we believe that seemingly low levels of P-selectin on sheared platelets are rather a snapshot of transient P-selectin exposure on platelets, followed by the rapid shedding of soluble fragment from the platelet surface. We have previously demonstrated that platelet exposure to CTD-generated shear stress for an extended time is associated with increased levels of circulating sP-selectin in vitro and ex vivo [23,52]. Increased levels of P-selectin on PDMPs also supports this hypothesis, suggesting that platelet degranulation occurs prior to or simultaneously with PDMP generation. Increased levels of P-selectin expression were previously reported for PDMPs released by platelets activated with thrombin [61]. Whether a similar mechanism of transient expression and subsequent shedding from the platelet surface is operative for PSGL-1 remains unclear.

The most well-known and therapeutically relevant *agonist receptors* are ADP-evoked P_2_Y_12_ and thrombin-evoked PAR_1_ receptors. Their surface expression is differentially regulated over the platelet lifespan correlating with platelet activation response [62]. We found that levels of these receptors on resting platelets were rather low; both P_2_Y_12_+ and PAR1+ platelet populations further decreased with increasing shear magnitude (Figure 7C,D). The surface density of P_2_Y_12_ on platelets remained unchanged, while PAR1 decreased slightly following shear. Similar to adhesion receptors, downregulation of agonist-evoked receptors on platelets may be a result of plasma membrane reorganization and receptor redistribution into sheared PDMPs. Indeed, small P_2_Y_12_+ and PAR1+ populations of sheared PDMPs were detected in our study (Figure 7A,B), and such a possibility was previously discussed by others [62]. Alternatively, it has been shown that platelet hyper-activation by biochemical agonists is associated with desensitization and internalization of P_2_Y_12_ and PAR1 receptors [63]. The internalized pool of these receptors was further tracked down to platelet endosomes and lysosomes [64]. We do not exclude the possibility that a small PDMP population carrying both P_2_Y_12_ and PAR1 receptors and showing up to 10-fold higher levels than sheared platelets may originate from platelet endosomes. Recent proteomic analysis of PDMPs released by resting and thrombin-activated platelets confirmed the presence of lysosomal proteins, specifically those regulating PAR1 internalization and recycling [65]. 

In earlier work, we demonstrated that sheared but not biochemically activated platelets expose negatively charged phospholipids and promote thrombin generation [17]. Here, we found that platelet exposure to mechanical forces, either shear or sonication, promotes externalization of phosphatidylserine on the platelets and facilitates shedding of numerous PDMPs with almost 3-fold higher density of negatively charged phospholipids than on platelets. In contrast, sonicated platelets and PDMPs demonstrated almost identical binding capacity for annexin V which was 6-fold lower than for sheared PDMPs. Sonication resulted in a much higher number of annexin V+ microparticles than those produced by shear. Taken together, these observations suggest that differing mechanisms of phosphatidylserine externalization are occurring in sheared and sonicated platelets. Specifically, increased phosphatidylserine surface density on sheared platelets and PDMPs is indicative of coordinated membrane lipid reorganization likely resulting from preceding signaling events and activation of platelet scramblases. Conversely, low-density annexin V binding on sonicated platelets and PDMPs may result from random platelet membrane damage, fragmentation, and formation of numerous microvesicles. 

Sheared and sonicated PDMPs demonstrated differing functional effects on thrombin generation and platelet aggregation. Sonicated PDMPs were more potent promoters of thrombin generation than sheared PDMPs when added alone or alongside resting or activated platelets (Figure 8D). We speculate that more abundant sonicated PDMPs cumulatively offer a larger net procoagulant surface, thus resulting in higher rates of thrombin generation as compared with the less abundant but more phosphatidylserine-dense sheared PDMPs. Moreover, demonstrating substantial prothrombinase activity, PDMPs may be more potent than platelets, even after short bursts of high shear stress with prothrombinase activity persisting and increasing during subsequent low-shear regimes [66]. When added to intact platelets in the plasma environment, both sheared and sonicated PDMPs inhibited platelet aggregation induced by collagen and ADP. Sheared PDMPs showed a slightly more pronounced anti-aggregatory effect than sonicated PDMPs (Figure 9). As such, our findings indicate that PDMPs generated by mechanical forces assume two antagonistic roles, being both procoagulant and anti-aggregatory. Densely covered with negatively charged phospholipids, PDMPs promote activation of prothrombin by factor Xa and generation of thrombin. As small and mobile membrane vesicles, PDMPs might also contribute to an acceleration of thrombosis and dissemination of the prothrombotic milieu from the bloodstream into organ tissues, thus contributing to microthrombosis associated with implantable CTDs [47,67]. On the other hand, we demonstrated that shear-generated PDMPs are also enriched with adhesion receptors (α_IIb_β_3_, GPIb-IX, GPVI, and PECAM-1) that are redistributed from platelets as a result of shear exposure. We do not exclude the possibility that PDMPs can compete with platelets for binding sites on the platelet surface and for adhesion protein ligands in plasma when platelets aggregate, thus effectively inhibiting aggregation. As such, shear-mediated generation of PDMPs, that show anti-aggregatory effect, and an associated decrease in adhesion receptors on platelets jointly contribute to device-related bleeding coagulopathy. Our in vitro findings are in alignment with the latest reports studying platelet function in CTD-supported patients. It was shown that the severity of acquired platelet defects, i.e., reduced GPIbα and GPVI expression and reduced α_IIb_β_3_ activation, is predictive of non-surgical bleeding events in CTD recipients, having even higher predictive power than vWF degradation parameters [37]. Further, device-implanted patients with increased bleeding risk demonstrate decreased platelet aggregation response, underscoring the importance of these acquired defects as drivers of device-related bleeding coagulopathy [68]. 

Regarding limitations, extensive sample processing is required for TEM, and it is difficult to distinguish circulating PDMPs from platelet filopodia or platelet fragments due to the apparent morphological similarity of these structures. Routine flow cytometry used to capture and characterize shear-mediated PDMPs is known to underestimate the small-size microparticle population (<300 nm) and exosomes (<100 nm), and such particles cannot be accurately sized using flow cytometry. Nonetheless, flow cytometry is an appropriate approach for identifying relative differences in platelet surface marker dynamics in response to shear. PDMP-rich fractions might contain soluble molecules released from platelets as a result of exposure to shear and sonication. To rule out the contribution of the soluble factors to observed anti-aggregatory and procoagulant effects of sheared and sonicated PDMPs, further research involving PDMP isolation is required. 

## 4. Methods

### 4.1. Blood Collection and Platelet Isolation

The study protocol was approved by the University of Arizona IRB (#1810013264). Blood was obtained via venipuncture from healthy volunteers and anticoagulated with acid citrate dextrose solution. Platelet-Rich Plasma (PRP) was obtained by centrifugation of anticoagulated blood at 400× *g* for 15 min. Gel-Filtered Platelets (GFP) were isolated from PRP by gel chromatography through Sepharose-2B [17]. Platelet fractions were stored and handled at room temperature to minimize storage-associated platelet function decline [25], apoptosis [44], and microparticle generation [69].

### 4.2. Platelet Exposure to Shear Stress and Sonication

Recalcified GFP (20,000 platelets/μL, 2.5 mM CaCl_2_) were subjected to uniform continuous shear stress (10–70 dyne/cm^2^, 10 min) in a hemodynamic shearing device [17,23,70]. Shear stress accumulation over time represents repeated platelet passages through high-shear regions of CTDs, selected based on previous numerical studies of CTD hemodynamics [71]. Alternatively, recalcified GFP were subjected to sonication for 10 s at 50% of power (Branson Ultrasonics™ SLPt Sonifier, Branson Ultrasonics, Brookfield, CT, USA).

To generate PDMPs, recalcified GFP (100,000 platelets/μL, 2.5 mM CaCl_2_) were subjected to 70 dyne/cm^2^ shear stress for 30 min. Alternatively, recalcified GFP were subjected to sonication. The sheared and sonicated samples were centrifuged twice at 2000× *g* for 10 min to sediment platelets [72]. The supernatant containing PDMPs was collected and stored on ice until used as a PDMP-rich fraction.

### 4.3. Platelet and Microparticle Imaging via Transmission Electron Microscopy

Following shear exposure or sonication, GFP (100,000 platelets/μL, 2.5 mM CaCl_2_) were fixed with 2.5% formaldehyde, 2.5% glutaraldehyde in 0.1 M sodium cacodylate buffer, pH 7.4, for 30 min and then centrifuged (1500× *g*, 15 min) to sediment intact platelets (platelet-rich low-speed fraction). The supernatant was collected and centrifuged (20,000× *g*, 30 min) to obtain the PDMP-rich pellet (high-speed fraction). The platelet- and PDMP-rich pellets were stored separately in 0.2 M sodium cacodylate buffer at 4 °C until TEM imaging [73]. Ultrathin sections (60 nm) were prepared as previously described [42] and examined via JEOL 1200EX (JEOL USA, Peabody, MA, USA) or FEI TecnaiG^2^ Spirit BioTWIN (FEI, Hillsboro, OR, USA) transmission electron microscopes at 80 kV. Images were captured with an AMT 2k CCD camera (Advanced Microscopy Techniques Corp., Danvers, MA, USA).

### 4.4. Surface Expression of Platelet Receptors on Platelets and Microparticles

Flow cytometric detection of platelet surface receptors was performed following published recommendations [69,74]. Sheared or sonicated GFP (20,000 platelets/μL, 2.5 mM CaCl_2_) were double-stained with five fluorescein-conjugated antibody pairs: anti-CD41-APC (Thermo Fisher Scientific, Waltham, MA, USA) & anti-PAR1-AF488 (R&D Systems, Minneapolis, MN, USA), anti-CD42a-FITC & anti-CD62P-APC (both from Thermo Fisher Scientific, Waltham, MA, USA), anti-CD31-FITC (Thermo Fisher Scientific, Waltham, MA, USA) & anti-PSGL1-PE (BD Biosciences, Franklin Lakes, NJ, USA), anti-CD41-APC & annexin V-FITC (both from Thermo Fisher Scientific, Waltham, MA, USA), or anti-GP6-PE (BD Biosciences, Franklin Lakes, NJ, USA) & anti-P_2_RY_12_-FITC (BioLegend, San Diego, CA, USA). GFP samples were stained for 30 min, fixed with 2% paraformaldehyde solution (Santa Cruz Biotechnology, Dallas, TX, USA) for 20 min, and diluted to 1 mL with Gibco^TM^ phosphate buffered saline (Thermo Fisher Scientific, Waltham, MA, USA). Flow cytometry was conducted on a FACSCanto II flow cytometer (BD Biosciences, Franklin Lakes, NJ, USA). Ten thousand events were captured within the stopping gate “Platelets + Microparticles”. FCS Express V3 software (DeNovo Software, Pasadena, CA, USA) was applied to analyze flow cytometry data. Single platelets were distinguished from microparticles based on their forward versus side scatter characteristics, as compared with standard polystyrene fluorescent nanobeads SPHERO^TM^ (Spherotech, Lake Forest, IL, USA) of 880 nm and 1350 nm [23]. Marker-positive platelet and microparticle counts were expressed as % of all events captured in a joint gate “Platelets + Microparticles”. Platelet and microparticle sizes were assessed as their median forward scatter [42,75]. The receptor arbitrary density on platelets and microparticles was calculated as the median fluorescence intensity normalized to the median forward scatter (MFI/MFS) [23,42]. 

### 4.5. Thrombin Generation on Platelets and Microparticles

The effect of platelets and PDMPs on prothrombin activation by factor Xa was assessed using the modified thrombin generation assay [76]. Following platelet exposure to shear stress, sonication, or PDMP-rich fraction (10 min, 37 °C), recalcified GFP (5000 platelets/μL, 5 mM CaCl_2_) were incubated with 200 nM acetylated prothrombin and 100 pM factor Xa (Enzyme Research Laboratories, South Bend, IN, USA) at 37 °C for 10 min. Then, 10 μL of each GFP sample were tested for thrombin activity in microplate wells containing 0.3 mM Chromozym TH (Roche Diagnostics, Indianapolis, IN, USA) and 3 mM EDTA. Kinetic changes in light absorbance (A405) were recorded for 7 min using a VersaMAX microplate reader (Molecular Devices, San Jose, CA, USA). The rate of thrombin generation was calculated as a slope of the kinetic curve.

### 4.6. Platelet Aggregation

A quantity of 300 µL of recalcified PRP (100,000 platelets/μL, 1 mM CaCl_2_) were incubated with 30 μL of PDMP-rich fraction or HEPES-modified Tyrode’s buffer containing 0.1% bovine serum albumin (for vehicle control) for 10 min at 37 °C with stirring. Platelet aggregation was initiated adding collagen (HELENA Laboratories, Beaumont, TX, USA), ADP (Sigma-Aldrich, St. Louis, MO), or TRAP-6 (TRAP-6, Roche Diagnostics GmbH, Mannheim, Germany). Alternatively, platelet aggregation was induced by the PDMP-rich fractions alone, and no biochemical agonist was added. Relative changes in light transmittance were recorded for 10 min by the optical aggregometer PAP-8E (Bio/Data Corp., Horsham, PA, USA). 

### 4.7. Statistical Analysis

Results from 4 to 7 independent experiments with different donors were summarized in the figures. All flow cytometry and aggregometry samples were run in duplicate. The data were tested for normality using the Shapiro–Wilk test and then statistically analyzed using one-way analysis of variance (ANOVA) followed by Dunnett’s multiple comparisons test and paired t-test for paired comparisons using GraphPad Prism 8 (GraphPad Software, San Diego, CA, USA). 

## 5. Conclusions

Our findings indicate that exposure to supraphysiologic shear stress accumulation promotes notable alterations in platelet morphology, including shape change, filopodia extension, intense degranulation, and generation of PDMPs. Shear-mediated microvesiculation is associated with differential remodeling of surface receptors expressed on platelets and microparticles, with several distinctive trends emerging. Sheared PDMPs impose a bidirectional effect on platelet hemostatic function, promoting thrombin generation and inhibiting agonist-induced platelet aggregation. Which mechanism prevails in vivo may vary in individual circumstances and remains to be defined. Although, it has become clear that shear-mediated PDMP generation is a selective and regulated process associated with the alteration of platelet morphology and function. From a translational perspective, our findings of heterogeneity in platelet response to shear and significant differences in PDMP phenotypes underscore that a range of operative mechanisms are at play in the platelet response to supraphysiologic shear imparted by CTDs. When coupled with the recognition of reduced efficacy of present antiplatelet agents as means of limiting CTD-related adverse events, this work supports the need to develop novel pharmacologic strategies that effectively target shear-mediated platelet dysfunction and generation of PDMPs. Successfully addressing these issues provides an opportunity for advancing the efficacy of CTDs while reducing patient risk.

## Figures and Tables

**Figure 1 ijms-24-07386-f001:**
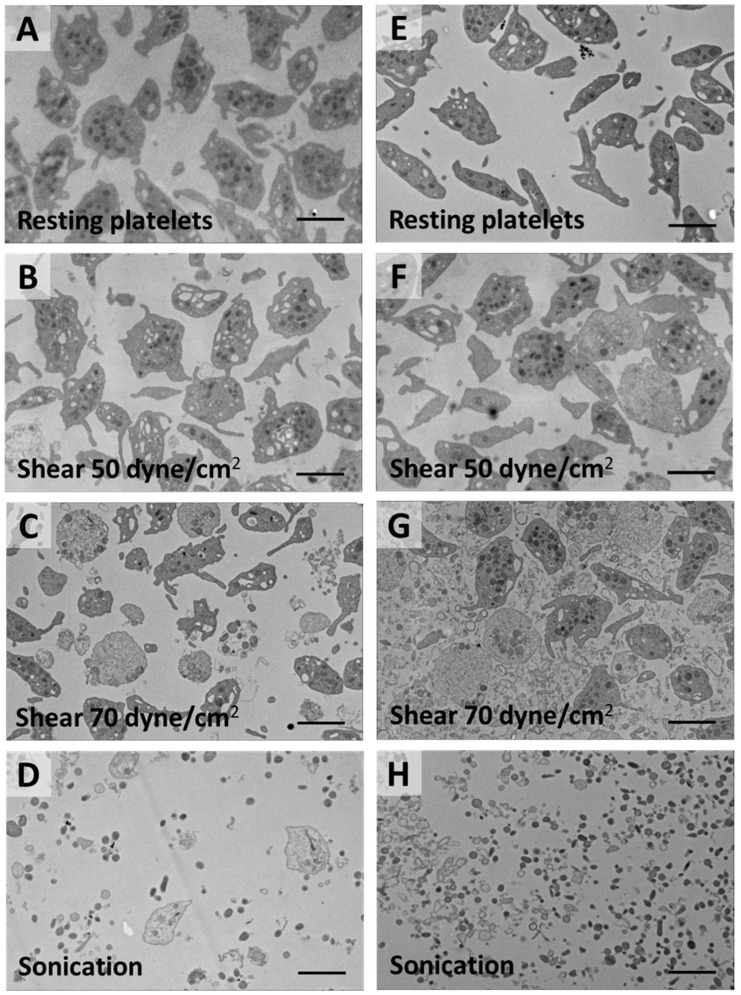
Shear stress of increasing magnitude and sonication promote alterations of platelet morphology and generation of platelet-derived microparticles. Illustrative transmission electron microscopy images of (**A**–**D**) platelets and (**E**–**H**) platelet-derived microparticles following exposure to shear stress (50 and 70 dyne/cm^2^) and sonication. (**A**–**D**) Platelets were pelleted via low-speed centrifugation (1500× *g*, 15 min). (**E**–**H**) Platelets and microparticles were pelleted by high-speed centrifugation (20,000× *g*, 30 min). Scale bar represents 2 μm. Direct magnification: 1200× (**A**), 2900× (**B**,**F**), and 3000× (**C**–**H**).

**Figure 2 ijms-24-07386-f002:**
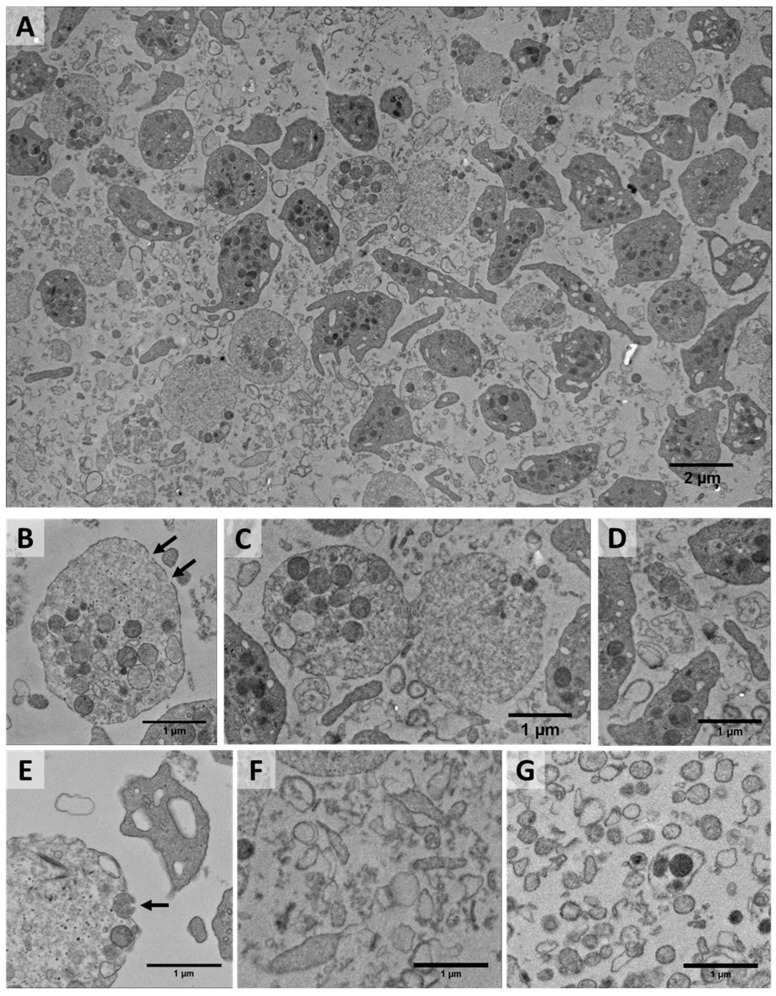
Illustrative thin-section electron microscopy images of platelets and platelet-derived microparticles (PDMPs) following exposure to 70 dyne/cm^2^ shear stress. (**A**)—morphological heterogeneity of platelets and PDMPs, (**B**,**C**)—platelet membrane thinning. (**D**)—large irregular particles, likely fragments of platelets or platelet structures, (**E**)—budding of platelet granules, (**F**,**G**)—small “granule-like” circular or ellipsoidal particles with evident contrast density or devoid of contents. Microparticles were pelleted by high-speed centrifugation (20,000× *g*, 30 min). The arrows indicate membrane thinning areas. The scale bar represents 2 μm (**A**) or 1 μm (**B**–**G**). Direct magnification: 3000× (**A**–**G**).

**Figure 3 ijms-24-07386-f003:**
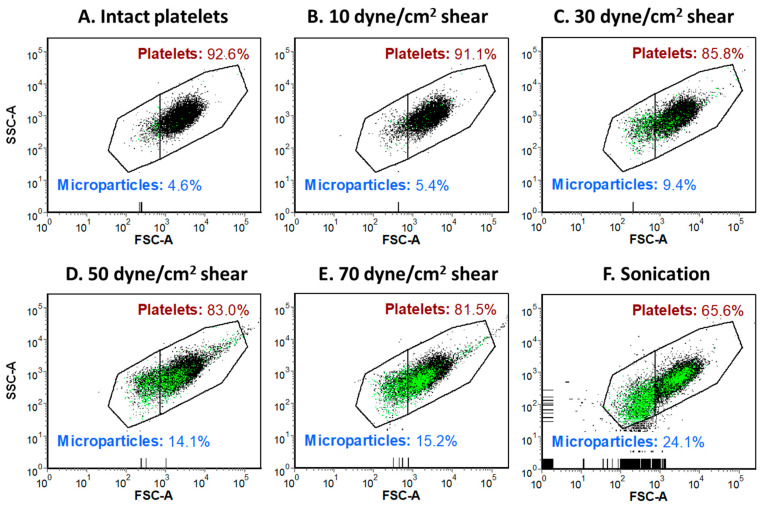
Platelet exposure to shear stress and sonication induces increased generation of platelet-derived microparticles (PDMPs). Illustrative dot diagrams of αIIb+ platelets and αIIb+ microparticles gated based on their forward and side scatter characteristics (**A**–**F**): (**A**)—intact platelets, (**B**–**E**)—sheared platelets, (**F**)—sonicated platelets. Black dots—CD41+ platelets or microparticles. Green dots—CD41+/annexin V+ double positive microparticles. FSC-A—forward scatter amplitude. SSC-A—side scatter amplitude.

**Figure 4 ijms-24-07386-f004:**
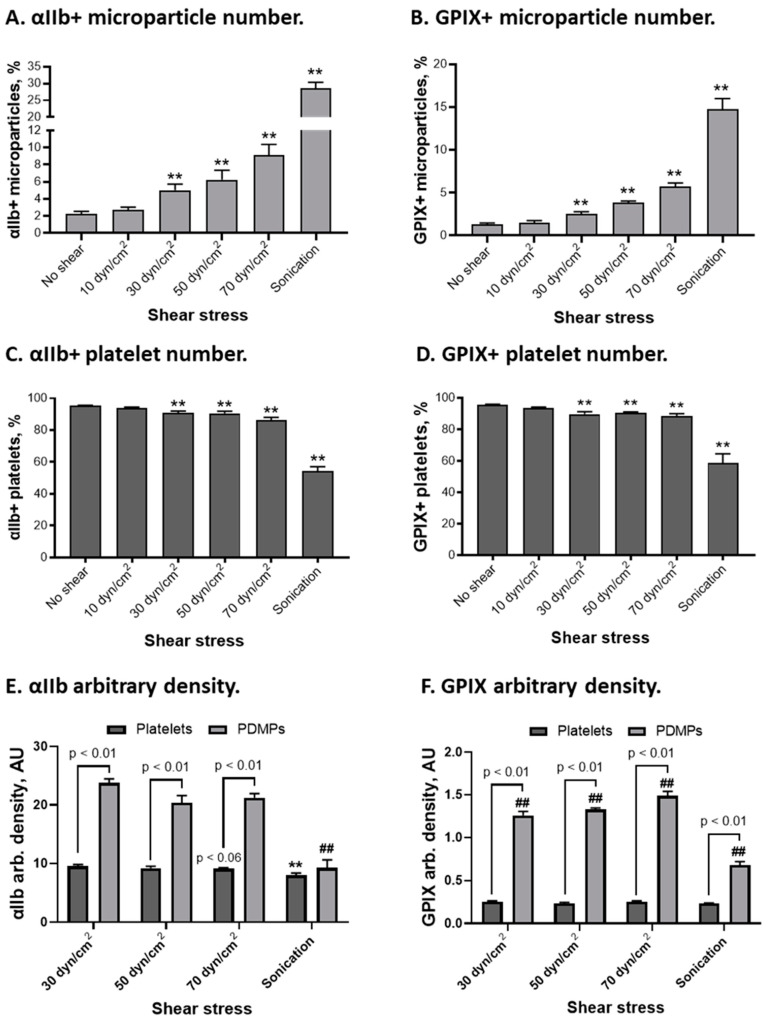
Distribution of αIIbβ3 integrin and GPIX on platelets and platelet-derived microparticles (PDMPs): (**A**,**B**)—αIIb (CD41)+ and GPIX (CD42a)+ PDMP number, (**C**,**D**)—αIIb (CD41)+ and GPIX (CD42a)+ platelet number, (**E**,**F**)—arbitrary density of αIIbβ3 (CD41) and GPIX (CD42) on platelets and PDMPs. *n* = 4–6. Mean ± SEM, 1-way ANOVA followed by Dunnett multiple comparisons test: **, ##—*p* < 0.01 vs. no shear for platelets and PDMPs.

**Figure 5 ijms-24-07386-f005:**
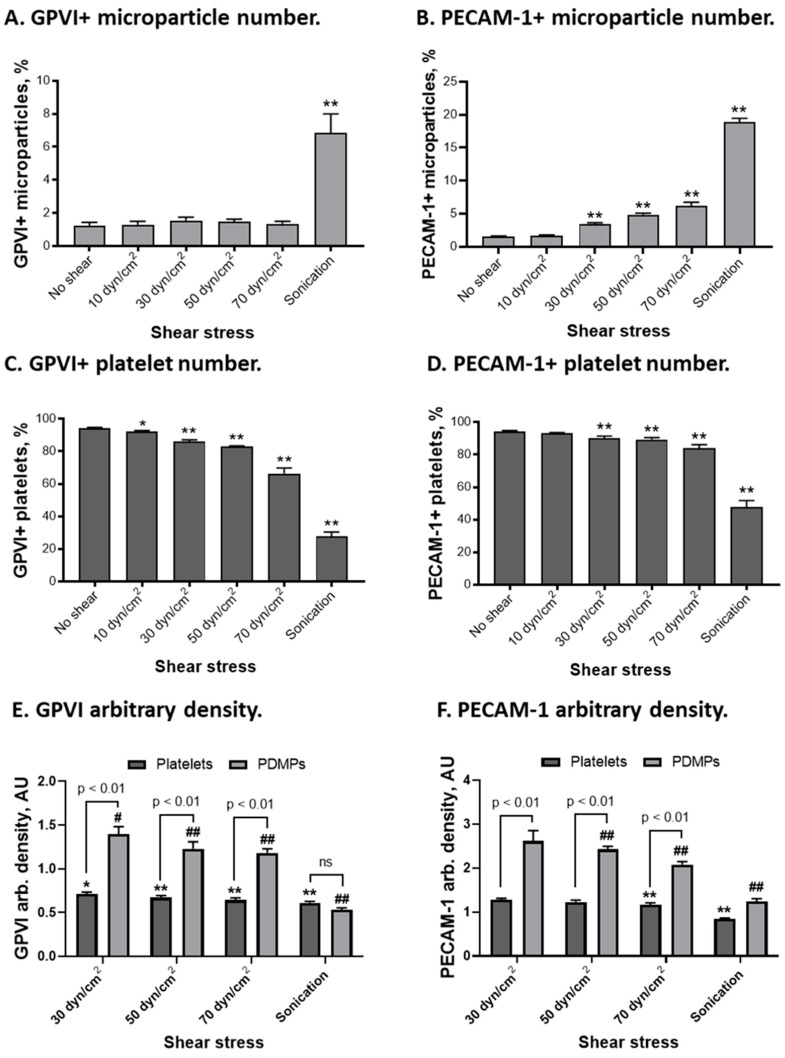
Distribution of GPVI and PECAM-1 receptors on platelets and platelet-derived microparticles (PDMPs): (**A**,**B**)—GPVI+ and PECAM-1+ PDMP number, (**C**,**D**)—GPVI+ and PECAM-1+ platelet number, (**E**,**F**)—arbitrary density of GPVI and PECAM-1 on platelets and PDMPs. *n* = 6–7. Mean ± SEM, 1-way ANOVA followed by Dunnett multiple comparisons test: *, #—*p* < 0.05 and **, ##—*p* < 0.01 vs. no shear for platelets and PDMPs.

**Figure 6 ijms-24-07386-f006:**
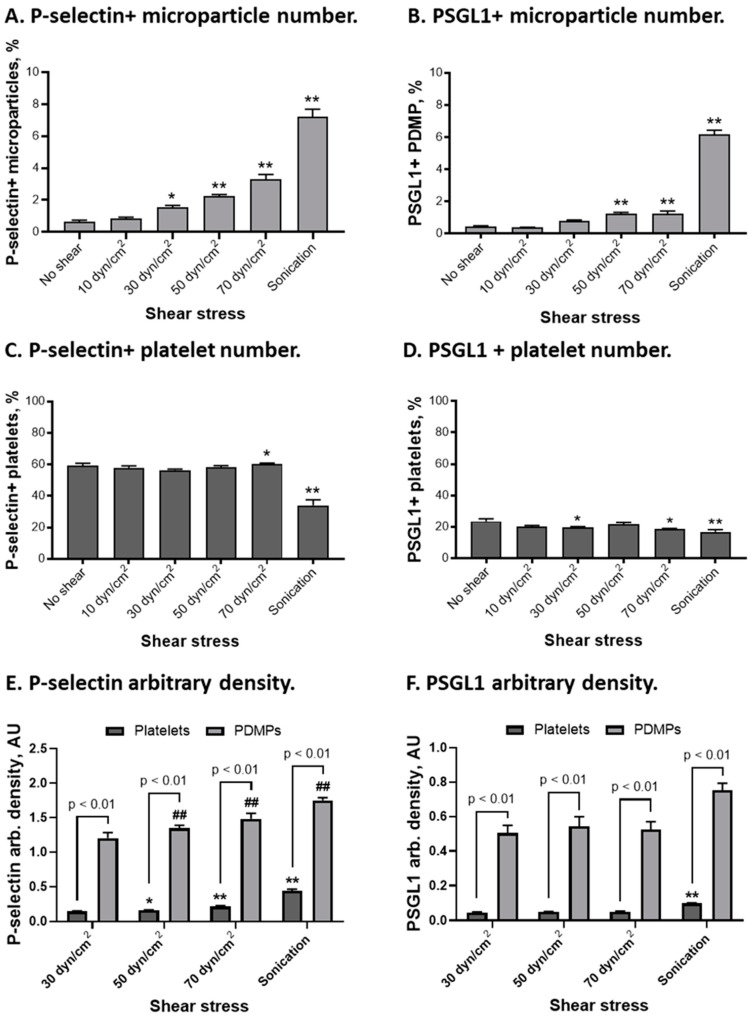
Distribution of P-selectin and PSGL1 on platelets and platelet-derived microparticles (PDMPs): (**A**,**B**)—number of P-selectin+ platelets and PDMPs (CD62+/CD42+ particles), (**C**,**D**)—number of PSGL1+ platelets and PDMPs (CD31+/PSGL1+ particles), (**E**,**F**)—arbitrary density of P-selectin and PSGL1 on platelets and PDMPs. *n* = 7. Mean ± SEM, 1-way ANOVA followed by Dunnett multiple comparisons test: *—*p* < 0.05 and **, ##—*p* < 0.01 vs. no shear for platelets and PDMPs.

**Figure 7 ijms-24-07386-f007:**
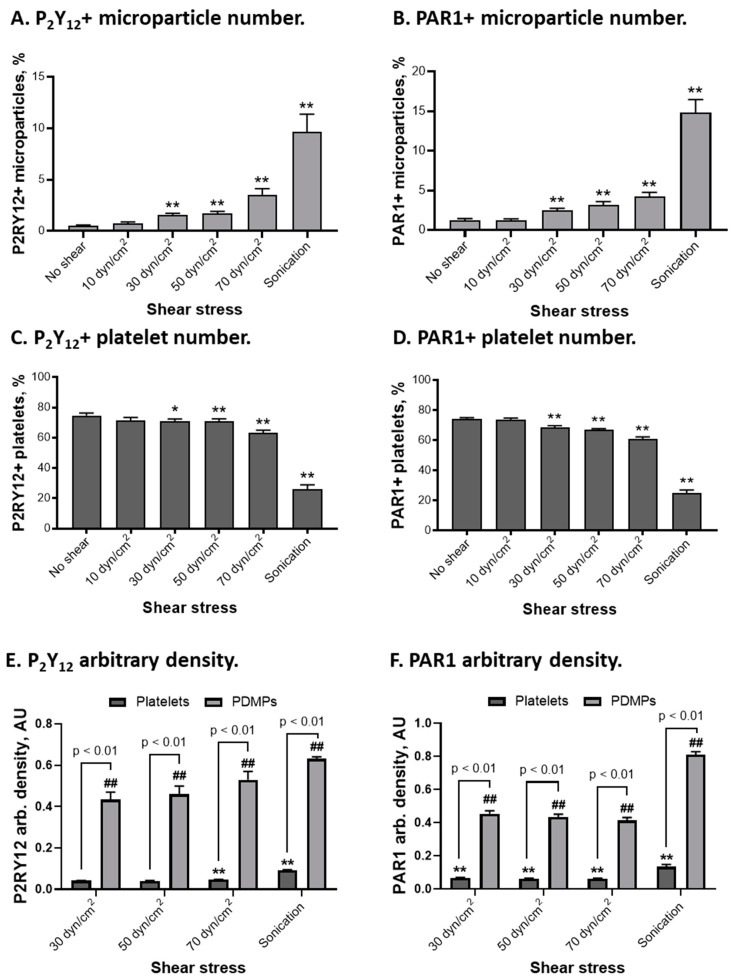
Distribution of P_2_Y_12_ and PAR1 on platelets and platelet-derived microparticles (PDMPs): (**A**,**B**)—number of P_2_Y_12_+ platelets and PDMPs (P2RY12+/ CD41+ particles), (**C**,**D**)—number of PAR1+ platelets and PDMPs (PAR1+/CD41+ particles), (**E**,**F**)—arbitrary density of P_2_Y_12_ and PAR1 on platelets and PDMPs. *n* = 4–7. Mean ± SEM, 1-way ANOVA followed by Dunnett multiple comparisons test: *—*p* < 0.05, **, ##—*p* < 0.01 vs. no shear for platelets and PDMPs.

**Figure 8 ijms-24-07386-f008:**
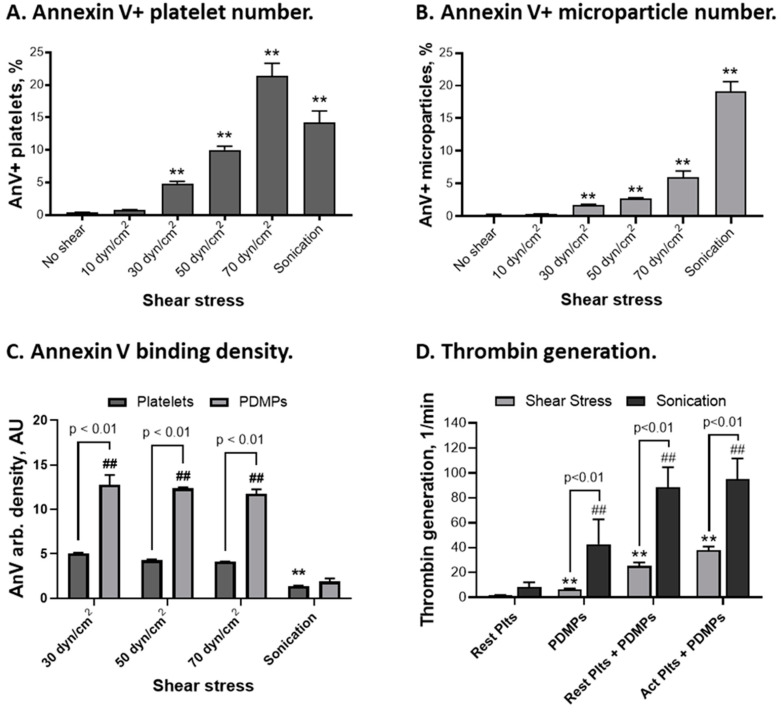
Platelets and platelet-derived microparticles (PDMPs) expose anionic phospholipids on their surface and promote thrombin generation exposure to shear stress and sonication: (**A**,**B**)—the number of platelets and PDMPs binding annexin V (AnV+/CD41+ particles), (**C**)—arbitrary density of annexin V binding on platelets and PDMPs. (**D**)—thrombin generation on platelets and microparticles generated via shear stress exposure (70 dyne/cm^2^, 30 min) or sonication: “Act Plts + PDMPs”—platelets and microparticles following shear exposure or sonication. *n* = 5–7. Mean ± SEM, 1-way ANOVA followed by Dunnett multiple comparisons test: **, ##—*p* < 0.01 vs. no shear for platelets and PDMPs.

**Figure 9 ijms-24-07386-f009:**
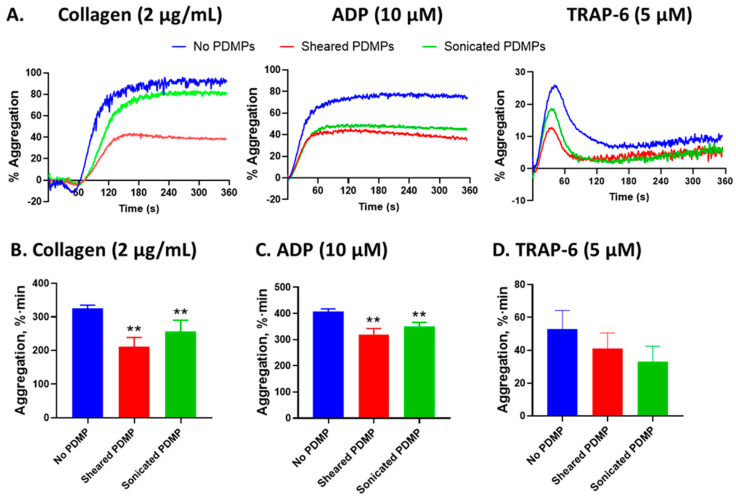
Platelet-derived microparticles (PDMPs) generated as a result of shear stress exposure and sonication inhibit agonist-induced platelet aggregation in plasma: (**A**)—representative aggregation curves; (**B**)—collagen-induced aggregation (*n* = 7–14), (**C**)—ADP-induced aggregation (*n* = 7–18), (**D**)—TRAP6-induced aggregation (*n* = 4–10). Mean ± SEM, 1-way ANOVA followed by Dunnett multiple comparisons test: **—*p* < 0.01 vs. no PDMP.

## Data Availability

The data presented in this study are available on request from the corresponding author.

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
