# Peer review of "Shear-Mediated Platelet Microparticles Demonstrate Phenotypic Heterogeneity as to Morphology, Receptor Distribution, and Hemostatic Function"

_ijms, 2023, doi:10.3390/ijms24087386_

Round 1

Reviewer 1 Report

This paper presents an experimental study on sheared platelet-derived microparticles (PDMPs). The authors test the hypothesis that sheared PDMPs manifest phenotypical heterogeneity of  morphology and receptor surface expression and modulate platelet hemostatic function.

The authors show that sheared PDMPs promote thrombin generation,  show an anti-aggregatory effect and are enriched with adhesion receptors. The authors also demonstrate that high shear stress results in notable alterations in platelet morphology. The findings agree with previous studies of increased levels of shear and suggest that PDMPs may be more potent than platelets, and  flow cytometry is an appropriate approach for identifying relative differences in surface marker dynamics in response to shear.

The paper is well-written, the quality of research is excellent, the results are novel and useful, and the topic is of interest to the readership of IJMS.

I recommend publication. However, the manuscript needs to be re-edited. There are big blanks on pages 5, 8, 10, 12, 14 and 16. Also, the margin on references is too narrow. They must be fixed before its publication.

Author Response

Dear Reviewer 1,

Thank you for your high regard for our manuscript. Please find our response below. 

Kindly,

Authors of the manuscript

Reviewer 2 Report

This is an interesting article examining the phenotypical heterogeneity of morphology and receptor surface expression on sheared PDMPs and how they may modulate platelet hemostatic function. This manuscript is a follow-up of the previous manuscript published by authors on Arterioscler Thromb Vasc Biol. 2021 April ; 41(4): 1319–1336. doi:10.1161/ATVBAHA.120.315583, and many platelet results were already reported in the first article.

The manuscript is well written, the research design is generally appropriate, and the results clearly presented. Overall, the observations are very interesting but the study has a big limitation: the authors used only the flow cytometry method to characterize PDMPs in terms of number and protein expression and to distinguish between large and small EVs.

Considerations:

1. the same authors wrote: “Routine flow cytometry used to capture and characterize shear-mediated PDMPs is known to underestimate the small size microparticle population (<300 nm) and exosomes (<100 nm), and such particles can-not be accurately sized using flow cytometry.” It would be better if the authors reported number and size of PDMPs also using a “conventional” method (eg. NTA).

2. It is not clear how the authors can distinguish small particles (150-500 nm) from large particles (500-1000 nm) using only the standard polystyrene fluorescent nanobeads SPHEROTM of 880 nm and 1350 nm (Spherotech). Why did the authors not use the appropriate SPHEROTM Nano Particle Size Kit composed of four bead sizes (0.22, 0.45, 0.88, and 1.33 mm) if the aim of that research was to analyse extracellular vesicles?

3. it is not clear on what basis the authors conclude that “flow cytometry is an appropriate approach for identifying relative differences in surface marker dynamics in response to shear.” It is well known that Flow cytometry underestimates small extracellular vesicles (<300 nm) and standard analyses are usually unable to characterize exosomes. Did the authors use a new and specific approach? If so, please include this very important information in the method section. Can they confirm some results using an ImagestreamX imaging FC, which provides visual confirmation of the detected particles and true differentiation from other smaller cells and aggregates, and it is able to detect more total MPs/ll than conventional flow cytometry?

4. Another very critical point is the effect of platelet-derived microparticles on platelet aggregation. How can the authors claim that the described effect on platelets is due only to PDMPs? Can they rule out that the effect here observed is not mediated by soluble factors present in the total supernatant (PDMP-rich fraction: method section line597-598)?

Minor.

Please include also the figure of large particles number

Author Response

Dear Reviewer 2,

Thank you for your review of the manuscript. Please find our point-by-point response to your suggestions below.

Kindly,

Authors of the manuscript

Round 2

Reviewer 2 Report

I thank the Authors for exhaustively answering all my questions.